# Effect of Zirconium Diboride and Titanium Diboride on the Structure and Properties of 316L Steel-Based Composites

**DOI:** 10.3390/ma16010439

**Published:** 2023-01-03

**Authors:** Iwona Sulima, Paweł Hyjek, Marcin Podsiadło, Sonia Boczkal

**Affiliations:** 1Institute of Technology, Pedagogical University of Krakow, Podchorazych 2 St., 30-084 Krakow, Poland; 2Łukasiewicz Research Network–Krakow Institute of Technology, Zakopianska 73 St., 30-418 Krakow, Poland; 3Łukasiewicz Research Network–Institute of Non-Ferrous Metals, Light Metals Division, Pilsudskiego 19 St., 32-050 Skawina, Poland

**Keywords:** titanium diboride, zirconium diboride, composites, sintering

## Abstract

The effect of zirconium diboride (ZrB_2_) and titanium diboride (TiB_2_) on the microstructure as well as the physical, mechanical, and tribological properties of composites based on 316 L steel is presented. Each reinforcing phase was added to the base alloy in the amount of 5 wt% and 10 wt%. The composites were fabricated by the SPS process (Spark Plasma Sintering). The results show that the weight fraction of the reinforcing phase affects the physical, mechanical, and tribological properties of the sintered composites. The sintered materials were characterized by a very high level of density. The addition of TiB_2_ has proved to be effective in increasing the hardness and compressive strength of the composites. The hardness of the composites with the addition of 10% TiB_2_ increased by 100% compared to the hardness of sintered 316L steel. It was found that introducing ZrB_2_ to the steel matrix significantly improved the wear resistance of the composites. The results showed that compared to 316L steel with the wear rate of 519 × 10^−6^ mm^3^/Nm, the wear rate of the composites containing 10% ZrB_2_ decreased more than twice, i.e., to 243 × 10^−6^ mm^3^/Nm.

## 1. Introduction

The development of Metal Matrix Composites (MMCs) is one of the main research areas. Owing to their unique properties, composites very often replace conventional materials in many engineering applications, especially under demanding operating conditions [1,2]. To improve the mechanical properties of composites, extensive research has been carried out, focused on the development of innovative and lightweight composite materials based on Ti, Al, Mg, and their alloys [3,4,5,6]. Alloys based on iron are also often used as a matrix of composite materials, offering both low-cost and high mechanical properties [7]. In the group of iron alloys, austenitic stainless steels are widely used, mainly because of the favorable combination of satisfactory mechanical properties and very high resistance to corrosion and oxidation [8,9]. Their main disadvantage is low hardness and wear resistance, which limits their use as a material for wear-resistant parts [10].

The important factors that determine the achievement of the optimum properties of composite materials include the choice of the proper reinforcing phase (material, content, and shape of precipitates), as well as the choice of the proper sintering process, parameters of the microstructure, and properties of sintered products (diffusion coefficient, viscosity, crystallographic structure, and presence of oxides) [11,12]. The use of TiB_2_ and ZrB_2_ ceramics is limited for technological reasons. Because of the high melting point, strong covalent bond, and low self-diffusion coefficient, it is difficult to obtain pure and compact boride ceramics using conventional sintering methods [13,14,15]. However, owing to their excellent properties, borides are preferred as the reinforcing phase of metal matrix composites. Borides such as ZrB_2_ and TiB_2_ are a special type of materials characterized by a high melting point (>3273 K), high hardness (ZrB_2_: ∼22–25 GPa, TiB_2_: ∼25–30 GPa) and Young’s modulus (ZrB_2_: 440–460 GPa, TiB_2_: 510–575 GPa), very high wear resistance, high oxidation and heat resistance, and satisfactory electrical properties [16,17,18]. Therefore, ZrB_2_ or TiB_2_ are increasingly used to reinforce steel matrix composites [19,20,21,22], aluminum alloys [23,24], copper alloys [25,26], and intermetallic phases [27].

The addition of boride ceramics improves not only the tribological properties of composites but also their stiffness and strength. Using the hot isostatic pressing process (HIP), Nahme et al. [28] sintered composite materials based on 316L steel reinforced with 15 vol% TiB_2_. It has been demonstrated that the use of fine ceramic particles increases the mechanical properties of the composite material. A uniform distribution of TiB_2_ in the steel matrix was obtained. For sintered 316L steel, the obtained values of Young’s modulus, yield strength, and tensile strength were 196 GPa, 225 MPa, and 520 MPa, respectively. In turn, for composites with 15% TiB_2_, an increase in the values of Young’s modulus (218 GPa), yield strength (495 MPa), and tensile strength (885 MPa) was observed. The research showed a decrease in the deformation from 45% for the 316L steel to 6% for the steel-15TiB_2_ composite. Wang et al. [29] generated in situ TiB_2_ particles in an Fe-Ti-B system using the SHS process (Self-propagating High-temperature Synthesis). It was demonstrated that the wear resistance of the TiB_2_-reinforced composite was higher than the corresponding wear resistance of the steel matrix. From the results of the research, it follows that, compared to composites reinforced with TiB_2_, the steel matrix was characterized by a double volume loss and a volume wear rate. Using hot isostatic pressing (HIP), Tjong and Lau [30] produced composites based on the steel matrix with 20 vol% TiB_2_. It has been shown that the addition of 20 vol% TiB_2_ increases the hardness from 148 HV1 (for 304 steel) to 264 HV1 (for composite). At the same time, it significantly reduces the volume loss of samples during abrasion tests. The wear resistance of the composites was over ten times higher than the wear resistance of the 304 steel without reinforcement. It was also observed that the volumetric wear of composites decreased with increasing load or speed.

Various methods are used to produce steel matrix composites, e.g., powder metallurgy (PM) [21,28,31,32,33], self-propagating high-temperature synthesis (SHS) [34], or conventional melting and casting [35]. The research described in [36,37,38,39] show that compared to free sintering methods, the use of the SPS technique allows producing finished sintered products, which are characterized by a high degree of compaction obtained in a shorter time at a lower pressure and temperature. In this study, an attempt was made to compare the effect of the addition of boride ceramics (TiB_2_ and ZrB_2_) on the properties and microstructure of sintered composites based on the austenitic 316L steel. The composites were sintered by the SPS method. The obtained materials were evaluated in terms of their microstructure as well as physical, mechanical, and functional properties. The results were compared with the 316L steel without reinforcement produced under the same sintering conditions.

## 2. Materials and Experimental Methods

To fabricate the composites, the powders of 316L steel (Fe-17.91Cr-12.03Ni-2.48Mo-0.23Mn-0.57Si-0.03C-0.02P-0.013S, in wt%; particle size of 25 µm, Hoganas, Sweden), diboride zirconium (ZrB_2_, 99.9 wt% purity, particle size of 2.5–3.5 μm, H.C. Starck, Germany) and diboride titanium (TiB_2_, 99.9 wt% purity, particle size of 2.5–3.5 μm, H.C. Starck, Germany) were used as the starting material. The microstructures of raw materials are shown in Figure 1.

ZrB_2_ and TiB_2_ powders were added to steel powders in an amount of 5 wt% and 10 wt%. The dry mixing of the composite powders was carried out in a Turbula model T2F (Willy A. Bachofen A, Muttenz, Switzerland) for 12 h with a rotation speed of 72 rpm using steel balls with a diameter of 5 mm. The sinters were produced by means of Spark Plasma Sintering (HP5 FCT System, Frankenblick, Germany) at 35 MPa. Disks of 20 mm diameter and 7 mm height were produced in graphite dies. Sintering was performed at a temperature of 1373 K in an argon atmosphere for 10 min. The applied heating and cooling rate was about 373 K/min. The temperature was monitored by a pyrometer. Figure 2 shows the selected sintering parameters that were registered during the SPS process.

After the sintering process, the relative density and porosity were measured by applying the Archimedes method using an analytical balance, RADWAG AS 220/C/2 (Radom, Poland), according to ASTM B962-08 [40]. The Young’s modulus was ultrasonically determined using the Panametrics Epoch III flaw detector (Billerica, MA, USA).

For metallographic examination, the surface of the sample was mechanically grounded and polished through standard procedures. The sinters were examined using scanning electron microscopy (Hitachi SU-70, Tokio, Japan) equipped with Wavelength Dispersive Spectroscopy (WDS). Furthermore, the microstructure and phase composition were studied using a high-resolution scanning electron microscope (INSPECT F50 FEI, Hillsboro, OR, USA) equipped with the EDAX OIM TSL EBSD system.

The Vickers hardness was evaluated using a hardness testing NEXUS 4000 (INNOVATEST EUROPE BV, Maastricht, The Netherlands) at a load of 2.942 N for 10 s. Each sample was tested by ten points. Compression tests were conducted on the INSTRON TT-DM machine (Norwood, MA, USA) at room temperature at a strain rate of 2.5 × 10^−4^ s^−1^. The tests were carried out on specimens with a height of 4.5 mm (h) and a diameter of 3 mm (d).

The wear resistance of sintered materials was examined using the ball-on-disc method, according to ASTM G99-05 standard [41]. Tests were performed on a universal testing machine manufactured by ELBIT (Koszyce Małe, Poland). Wear tests were conducted in a dry environment at ambient temperature using an alumina ball (r—1/8″) as the sliding counter body. In wear tests, the following parameters were applied: radius of the friction track—5 mm, load—5 N, test duration t = 10,000 s, total sliding distance—1000 m, and sliding speed—0.2 m/s. During the wear tests, coefficients of friction (µ) for the contact with the Al_2_O_3_ ball, wear scar depth, and specific wear rate (W_v_) were specified. The specific wear rate was calculated from the V_disc_/F_n_L equation, where V_disc_ is the wear volume of the disc specimen, F_n_ is the applied normal load, and L is the sliding distance [41]. In the next step, the worn surfaces were investigated using SEM (JEOL JSM 6610LV, Tokyo, Japan).

## 3. Results and Discussion

The densification data of the composites with ZrB_2_ and TiB_2_ sintered at the temperature of 1373 K with 10 min of holding time are shown in Table 1. The relative density was calculated as the ratio of the measured density to the calculated theoretical density of the composite [31]. The relative densities of the sintered composites are in the range of 96–98%, which confirms the high degree of compaction. However, it can be seen that the measured density values (Table 1) are lower than the theoretical values, which indicates that some porosity was present in the composite samples. After the SPS process, the sinters were characterized by a porosity of less than 1.5%. The applied sintering conditions allowed for a reduction in porosity. The results show that the introduction of TiB_2_ or ZrB_2_ ceramics into the steel matrix did not deteriorate the density and porosity of the composites. With the increasing content of the reinforcing phase, the density was gradually decreasing. For sintered steel, the density was 7.85 g/cm^3^. In turn, for composites containing 10% TiB_2_ or 10% ZrB_2_, the density was 7.39 g/cm^3^ or 7.29 g/cm^3^, respectively. These results can be compared with the results obtained by Sahoo et al. [42], who produced, by hot isostatic pressing (HIP), a composite material based on the steel matrix reinforced with 2–4 vol% of TiB_2_ particles. They also demonstrated that the density of that material decreased with the increasing content of the ceramic phase. The values of the relative density were in the range of 87–92%.

Figure 3 shows the microstructure of the composites as a function of the type of reinforcement and its weight fraction. Comparing the microstructures, it was found that borides tended to be located along the grain boundaries of the steel matrix. The distribution of the reinforcing phase was basically even for the steel + ZrB_2_ and steel + TiB_2_ composites. Only in the case of composites containing 10 wt% ZrB_2_ (Figure 3d), the formation of agglomerations of the ceramic phase was observed locally, which may be the result of insufficient preparation of the composite powder mixture at the mixing stage in Turbula. In the WDS examinations of the chemical composition of the steel+ZrB_2_ and steel+TiB_2_ composites (Figure 4 and Figure 5), the presence of very fine chromium-containing precipitates was detected. The results of the EBSD analysis are compared in Figure 6 and Figure 7, respectively. The phase analysis of the steel+ZrB_2_ composites showed the presence of the following phases in the microstructure: Fe, ZrB_2_, and a chromium-containing phase. In turn, for the steel+TiB_2_ composites, the following phases were detected in the microstructure: Fe, TiB_2_, and a chromium-containing phase.

In Figure 8 and Table 1, the results of Young’s modulus tests are given. The analysis of the data showed that the Young’s modulus increased with the increasing content of the reinforcing phase. The highest values were obtained for composites containing 10 wt% of the reinforcing phase. It was observed that the addition of ZrB_2_ to the steel matrix gave higher values of the Young’s modulus than the addition of TiB_2_. For composites reinforced with 5% and 10% TiB_2_, the values of the Young’s modulus were 205 GPa and 216 GPa, respectively, while 5% and 10% ZrB_2_ added to the steel matrix produced the Young’s modulus values of 211 GPa and 227 GPa, respectively. For comparison, the Young’s modulus of sintered 316L steel was 198 GPa.

The mechanical properties of the sintered materials are shown in Figure 9 and Figure 10. It was observed that the hardness of the composites increased with the increasing weight fraction of ceramics in the steel matrix. For sintered 316 L steel, the hardness was 214 HV0.3. The addition of 5 wt% of the reinforcing phase increased the hardness by about 40% and 65% for ZrB_2_ and TiB_2_, respectively. It was found that composites reinforced with TiB_2_ had higher hardness values than composites containing ZrB_2_. This is clearly visible in the composites with 10 wt% of the reinforcing phase. The hardness of the composites with the addition of 10% TiB_2_ more than doubled compared to the hardness of 316L steel without reinforcement. For comparison, the hardness of the composites with the addition of 10% ZrB_2_ increased by about 70% compared to the hardness of 316 L steel without reinforcement. In an attempt to better assess the mechanical properties, room-temperature compression tests were performed under conditions of uniaxial compressive loading. The analysis of the stress–strain curves (Figure 10) showed that the addition of a ceramic phase to the steel matrix had improved the compressive strength of the tested composites. Generally speaking, TiB_2_ had a more favorable effect on the compressive strength of the composites. Compared to sintered 316L steel, the highest increase in strength of about 60% was obtained for composites with 10% TiB_2_. In contrast, the addition of 10% ZrB_2_ increased the compressive strength by about 30%. On the other hand, the plasticity of these composites was lower than the plasticity of sintered steel. Similar results were obtained in other studies, where the phase reinforcing the steel matrix was composed of oxides (Y_2_O_3_) and carbides (TiC) [43,44]. The conducted studies also showed that changing the content of the TiB_2_ or ZrB_2_ reinforcing phase had only an insignificant effect on the compressive strength of the composites (Figure 11). The improvement in the mechanical properties of sintered composites was due to the introduction of a fine reinforcing phase and its uniform distribution in the steel matrix, observed in all steel-TiB_2_ composites (Figure 3). Additionally, small chromium-containing precipitates, also acting as a reinforcement of the microstructure, were formed during the SPS process in the composite microstructure (Figure 4, Figure 5, Figure 6 and Figure 7). The uniform distribution of the ceramic reinforcing phase and the presence of fine chromium precipitates are a barrier to dislocations and result in their accumulation during deformation. The increase in the strength of sintered composites can be attributed to strain hardening by the Orowan mechanism [42,45,46].

The wear resistance of the composites was assessed by measurements of the coefficient of friction and the determination of the specific wear rate in a ball-on-disc test. Figure 11 shows the effect of the content of two ceramic additives on the tribological properties of composites. From the obtained results, it follows that the coefficient of friction and the specific wear rate depend on the amount, as well as the type of boride phase. The coefficient of friction and the specific wear rate decrease with the reinforcing phase content increasing in the steel matrix. The additions of ZrB_2_ and TiB_2_ improved the wear resistance of austenitic stainless steel, owing to the high hardness of the ceramic particles [17,18], which hinders the wear of the reinforcement and, at the same time, protects the steel matrix from intensive wear. Moreover, the results have shown that the use of ZrB_2_ allowed for obtaining better tribological properties. Regardless of the content of the reinforcing phase, the coefficient of friction and the specific wear rate were always lower when ZrB_2_ was used.

The coefficient of friction of sintered steel was 0.64. On the other hand, for composites containing 10 wt% of the reinforcing phase, the lowest values of the coefficient of friction were obtained, i.e., equal to 0.43 and 0.39 for TiB_2_ and ZrB_2_, respectively (Figure 11a). A similar trend was observed for the results of the calculations of the specific wear rate (Figure 11b). This is attributed to the fact that the reinforcing phases improve both hardness and strength of the composites and provide a hard barrier to matrix deformation and wear [47,48,49]. During abrasion tests, the 316L steel showed a higher degree of wear. The specific wear rate of steel was 519 × 10^−6^ mm^3^/Nm. The addition of ceramic particles reduced the specific wear rate of the composite, reducing the abrasion rate nearly two times. As a result, for composites containing 10 wt% of the reinforcing phase, the specific wear rate was 285 × 10^−6^ mm^3^/Nm and 243 × 10^−6^ mm^3^/Nm for TiB_2_ and ZrB_2_, respectively. This is in line with Tjong’s research [30], which demonstrated a significant improvement in the wear resistance of composites with the increasing content of TiB_2_. Dinahran and Murugan [50] also found in their studies that the specific wear rate of aluminum alloy composites was gradually decreasing with the increasing percentage content of ZrB_2_. From the results of the research, it follows that the wear rate of the composites undergoes a linear improvement with the increase in normal load, sliding velocity, and sliding distance.

Figure 12 shows the morphology (SEM) of the wear track. In the case of the steel sample, the contact surface suffered heavy wear (Figure 12a). The dominant wear mechanism was abrasive wear. The worn surface was characterized by numerous deep scratches and furrows combined with plastic deformation. Flat areas were observed in the wear zone, which can be attributed to strong plastic deformation of the soft steel matrix under normal load. In contrast, the worn surfaces of composite samples were smoother, and the grooves and furrows were fine. This indicates that the addition of the reinforcing phase increased the hardness of the composite, leading to improved wear resistance. Figure 13 shows the microstructure of the surface of the Al_2_O_3_ ball after the tribological test. After each test, the ball surface exposed to wear was damaged and rough with visible scratches. This indicates that the Al_2_O_3_ ball wear process occurred during contact with the composite surface.

## 4. Conclusions

The effect of boride ceramics (TiB_2_ or ZrB_2_) on the properties and microstructure of 316 L steel matrix composites sintered by the SPS process was examined. From the obtained results, the following conclusions can be drawn:In all tested steel+ZrB_2_ and steel+TiB_2_ composites, a very high degree of compaction was achieved. Relative densities of sintered composites were in the range of 96–98%.The properties of sintered composites are closely related to the weight fraction and the type of boride phase.Young’s modulus and hardness increase with the reinforcing phase content increasing in the steel matrix. It was also demonstrated that the coefficient of friction and the specific wear rate decrease with the increasing content of TiB_2_ or ZrB_2_.Compared to composites containing ZrB_2_, composites reinforced with TiB_2_ are characterized by higher mechanical properties.In contrast to the 316 L steel without reinforcement, the hardness and the compressive strength of the composites with the addition of 10% TiB_2_ increased by 100% and 60%, respectively. On the other hand, the addition of ZrB_2_ ceramics to the steel matrix improved the wear resistance of the composites. The lowest values of the coefficient of friction and the specific wear rate were obtained in the steel+10% ZrB_2_ composites.

## Figures and Tables

**Figure 1 materials-16-00439-f001:**
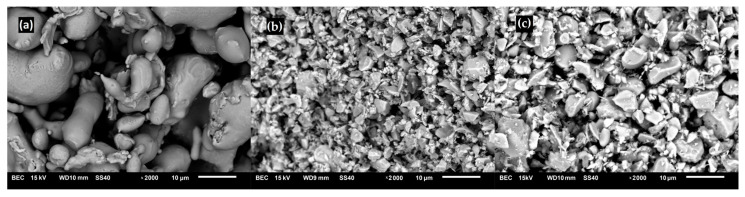
SEM images of raw materials. (**a**) 316L austenitic stainless steel, (**b**) diboride zirconium, and (**c**) diboride titanium.

**Figure 2 materials-16-00439-f002:**
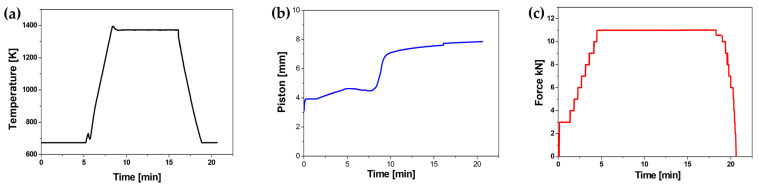
Actual (**a**) temperature–time, (**b**) displacement–time, and (**c**) pressure–time curves registered during the SPS process.

**Figure 3 materials-16-00439-f003:**
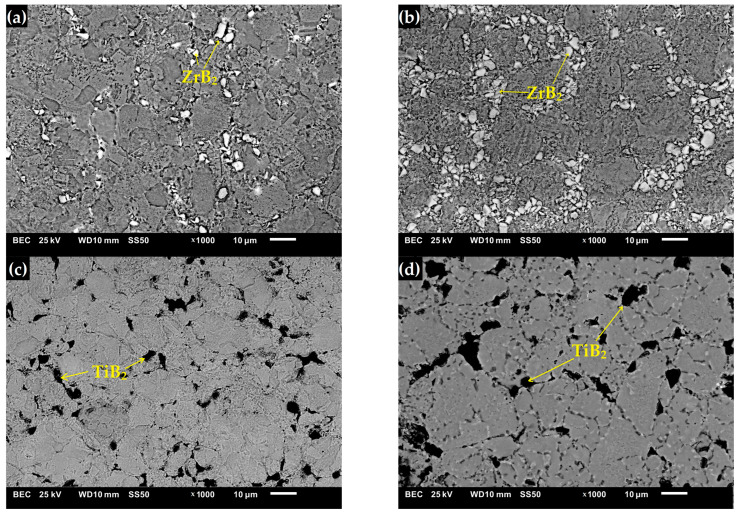
The SEM micrograph of the composites with: (**a**) 5% ZrB_2_, (**b**) 10% ZrB_2,_ (**c**) 5% TiB_2_, and (**d**) 10% TiB_2_.

**Figure 4 materials-16-00439-f004:**
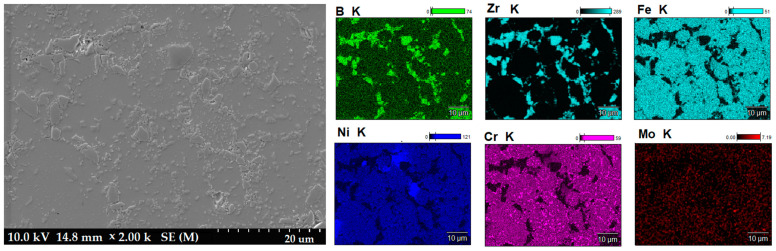
The SEM image of steel+10% ZrB_2_ composite with WDS analysis.

**Figure 5 materials-16-00439-f005:**
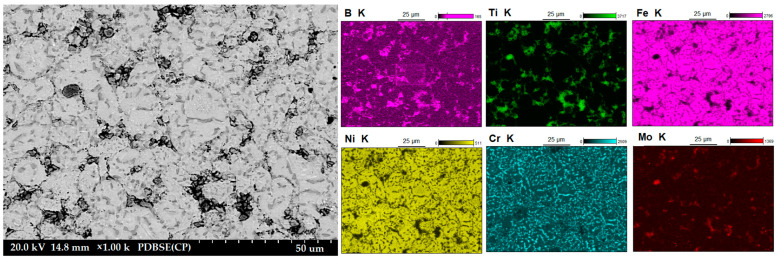
The SEM image of steel+10% TiB_2_ composite with WDS analysis.

**Figure 6 materials-16-00439-f006:**
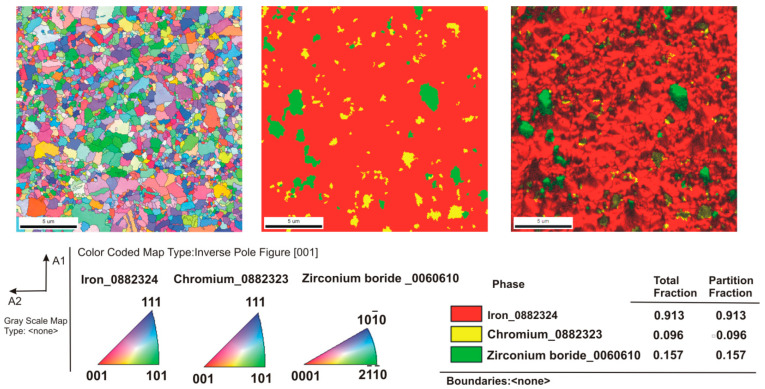
The EBSD maps of the steel+5% ZrB_2_ composite.

**Figure 7 materials-16-00439-f007:**
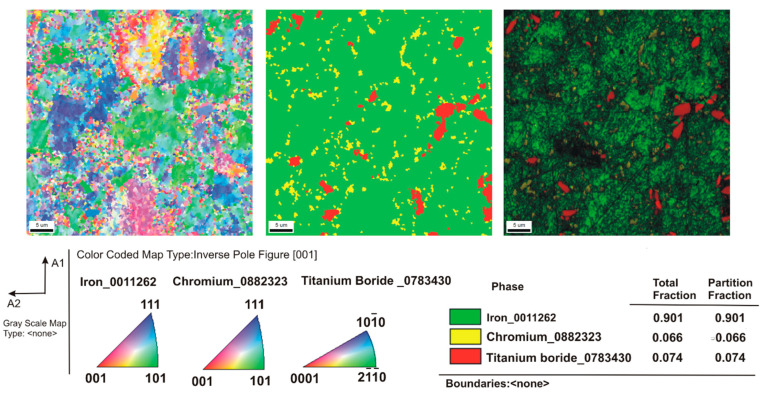
The EBSD maps of the analysis of the steel + 5% TiB_2_ composite.

**Figure 8 materials-16-00439-f008:**
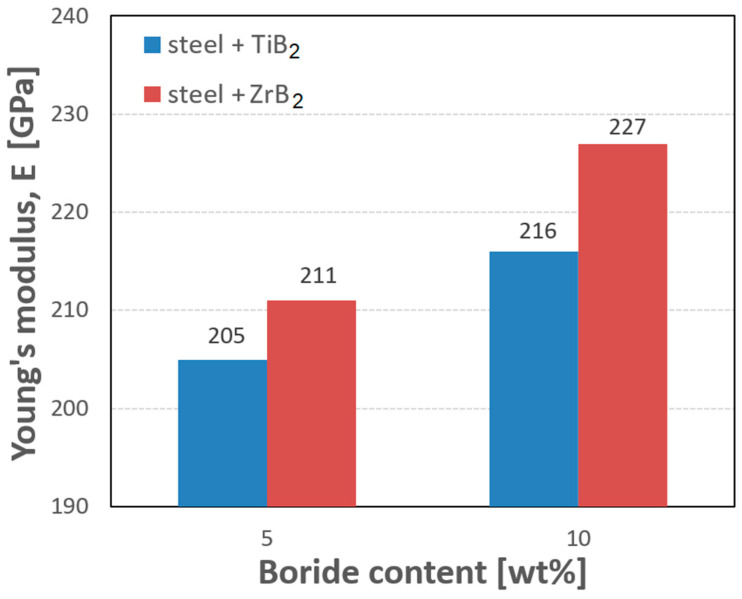
The results of the Young’s modulus measurements.

**Figure 9 materials-16-00439-f009:**
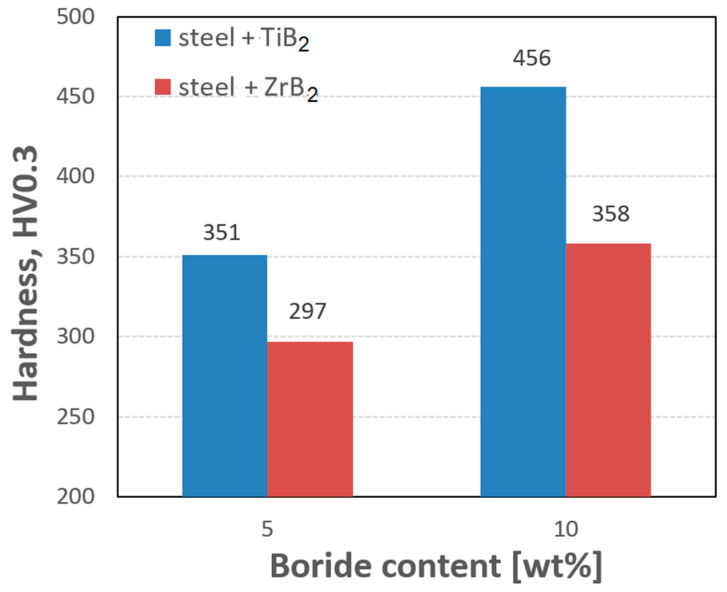
Vickers hardness of sintered composites.

**Figure 10 materials-16-00439-f010:**
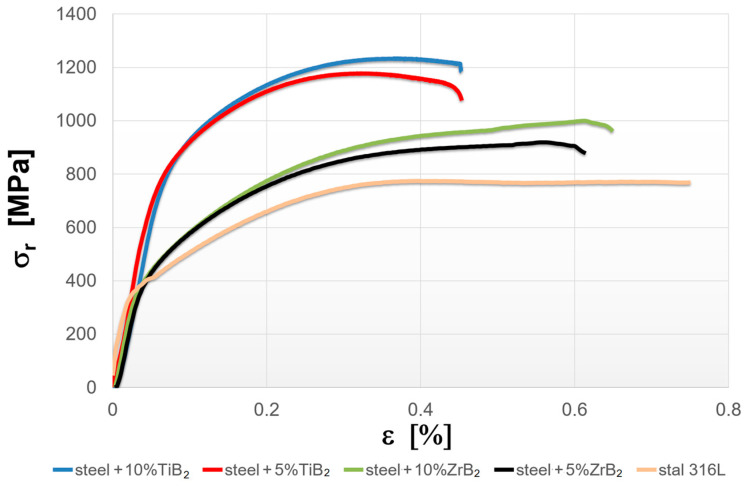
Compressive stress–strain curves for sintered materials.

**Figure 11 materials-16-00439-f011:**
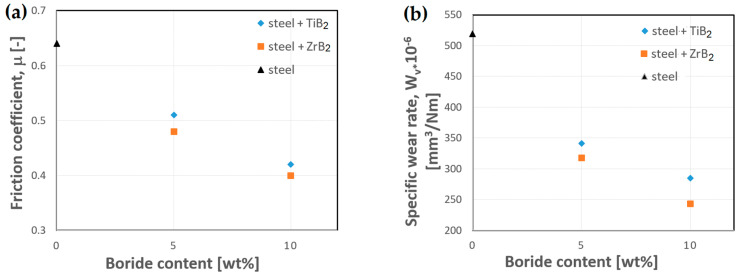
Variations in (**a**) the coefficient of friction and (**b**) the specific wear rate as a function of the content of ceramic compositions.

**Figure 12 materials-16-00439-f012:**
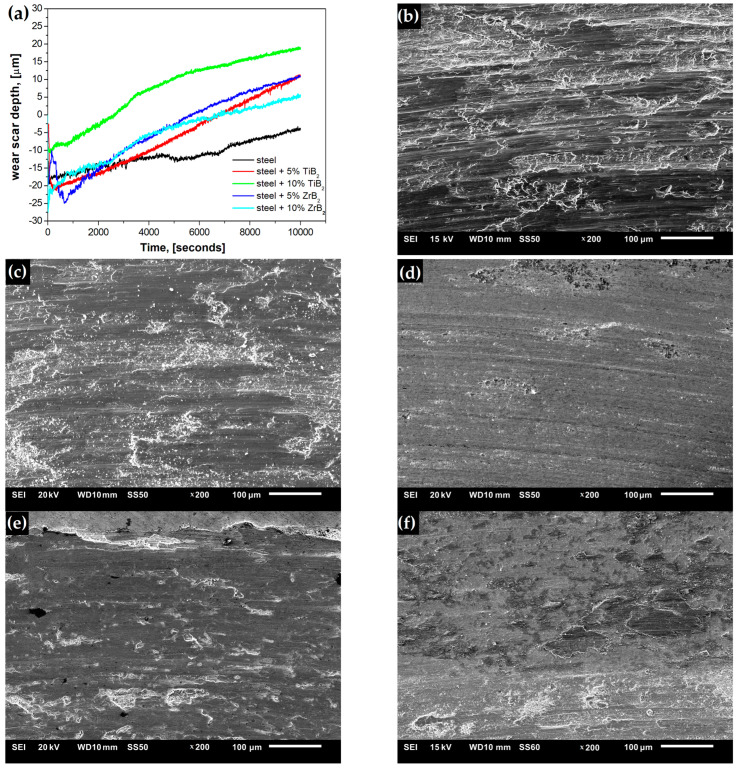
(**a**) Depth of scar as a function of the test duration and SEM micrograph of the worn surface of: (**b**) steel and composites with (**c**) 5% ZrB_2_, (**d**) 10% ZrB_2_, (**e**) 5% TiB_2_, and (**f**) 10% TiB_2_.

**Figure 13 materials-16-00439-f013:**
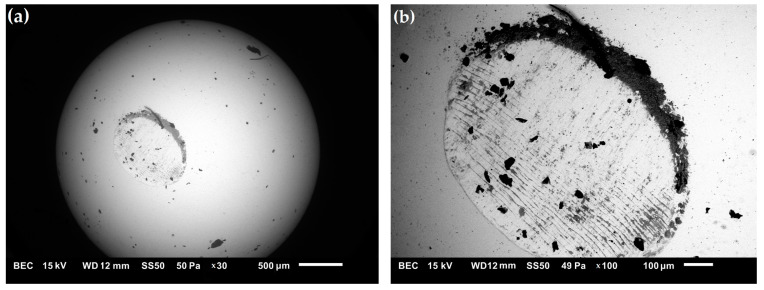
(**a,b**) SEM micrograph of the wear surface of the Al_2_O_3_ ball after the tribological test.

**Table 1 materials-16-00439-t001:** Properties of the materials formed by SPS.

Sintered Materials	Density (g/cm^3^)	Relative Density (%)	Porosity (%)	Young’s Modulus (GPa)	Relative Young’s Modulus (%)
steel	7.85	98	0.6	198	95
steel + 5% ZrB_2_	7.52	96	1.1	211	94
steel + 10% ZrB_2_	7.29	96	1.5	227	95
steel + 5% TiB_2_	7.61	97	0.9	205	91
steel + 10% TiB_2_	7.39	96	1.3	216	92

## Data Availability

The data presented in this study are available on request from the corresponding author.

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
