# Peer review of "Effect of Zirconium Diboride and Titanium Diboride on the Structure and Properties of 316L Steel-Based Composites"

_materials, 2023, doi:10.3390/ma16010439_

Round 1

Reviewer 1 Report

The particle size of the TiB2 powder 2.5-3.5 micrometers in the text does not correspondent to Figure 1c (up to 10 micrometers)

Author Response

Thank you for  attention. Figure 1c has been changed in the article.

Reviewer 2 Report

Dear Authority,

The manuscript entitled ‘Role of the zirconium diboride and titanium diboride on the structure and properties of metal matrix composites’ presents the structural characterization of zirconium diboride and titanium diboride reinforced 316L composites. I think, the paper presents valuable information about improvement on mechanical properties such as hardness and compressive strength along with tribological features. The effect of reinforcement species in 316L steel is properly explained. However, there is missing information in abstract needs improvement. In manuscript body, the data acquired in this study needs to link with some theories about mechanical alloying system. The manuscript needs to be revisited by considering following comments;

1- Abstract is not good, improve it with the numerical results and write it again succinctly. The main theme of this paper is not described in the abstract. Abstract section should be concisely reflected the content and summarize the problem, the method, the results, and the conclusions. For instance, the important information which is 100% hardness improvement for TiB2 reinforced alloy, could be added in abstract part.

2- Enhancement in mechanical properties could be explained with the mechanism such as excessive plastic deformation, increased dislocation density, the restriction of dislocation mobility, Orowan mechanism etc arising with high energy ball milling process. please refer following manuscripts;

a)  https://doi.org/10.1016/j.mtcomm.2021.102202

b) https://doi.org/10.1016/j.mtcomm.2021.102637

After minor modification, the paper could be considered for publication in Materials.

Best wishes,

Author Response

The article was again reviewed by a native speaker.

1- Abstract is not good, improve it with the numerical results and write it again succinctly. The main theme of this paper is not described in the abstract. Abstract section should be concisely reflected the content and summarize the problem, the method, the results, and the conclusions. For instance, the important information which is 100% hardness improvement for TiB2 reinforced alloy, could be added in abstract part.

Abstract was corrected.

2- Enhancement in mechanical properties could be explained with the mechanism such as excessive plastic deformation, increased dislocation density, the restriction of dislocation mobility, Orowan mechanism etc arising with high energy ball milling process. please refer following manuscripts;

  1. a)  https://doi.org/10.1016/j.mtcomm.2021.102202
  2. b) https://doi.org/10.1016/j.mtcomm.2021.102637

The authors thank you very much for your valuable comments and interesting publication, which were used in our publication. Currently, the authors are conducting detailed studies of the mechanical properties of:

  • tensile tests at temperatures ranging from 20 to 900°C
  • compression tests at temperatures ranging from 400 to 900°C
  • detailed microstructure characteristics (SEM, TEM, fractography) before and after the mechanical tests

The results of these studies will be published in a separate article. Also, the aim of these studies will be to analyze the mechanisms influencing the change in the mechanical properties of composites.

Reviewer 3 Report

The reviewed article titled "Role of the zirconium diboride and titanium diboride on the structure and properties of metal matrix composites" needs a serious rework before it can be recommended for publication. The English language of the manuscript needs serious editing. It is recommended that the authors refer the article to professional English language editing services if possible. Although the results of the work are not novel, they are still worth publishing. I strongly advise the authors to carefully review the manuscript once again. I have attached my comments in a separate PDF file.

Author Response

The authors thank you very much for for valuable comments. The text of the manuscript has been revised according to the comments. The article was again reviewed by a native speaker.  I have attached my answers in a separate file.

Round 2

Reviewer 3 Report

The authors addressed all questions extensively. I recommend the paper for publication in its present form.